# Structured and Abstractive Reasoning on Multi-modal Relational Knowledge Images

## Abstract

Understanding and reasoning with abstractive information from the visual modality presents significant challenges for current multi-modal large language models (MLLMs). Among the various forms of abstractive information, **M**ulti-**M**odal **R**elational **K**nowledge (MMRK), which represents abstract relational structures between multi-modal entities using node-edge formats, remains largely underexplored. In particular, **ST**ructured and **A**bstractive **R**easoning (STAR) on such data has received little attention from the research community. To bridge the dual gaps in large-scale high-quality data and capability enhancement methodologies, this paper makes the following key contributions: (i). An automatic STAR data engine capable of synthesizing images with MMRK to build multi-modal instruction data with reliable chain-of-thought thinking for various STAR tasks and (ii). A comprehsive two-stage capability enhancement training framework, accompanied by a suite of evaluation protocols tailored to different STAR tasks. Based upon these contributions, we introduce STAR-64K, a dataset comprising 64K high-quality multi-modal instruction samples, and conduct experiments across 5 open-source MLLMs. Experimental results show that our two-stage enhancement framework enables smaller 3B/7B models to significantly outperform GPT-4o in STAR. Additionally, we provide in-depth analysis regarding the effectiveness of various designs, data transferability, and scalability.

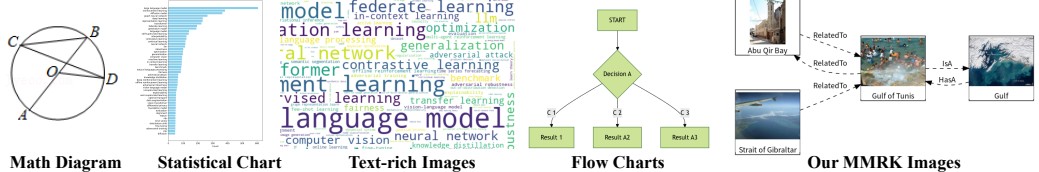

Math Diagram    Statistical Chart    Text-rich Images    Flow Charts    Our MMRK Images

Figure 1: Different kinds of images contain abstractive information with complex semantics.

## 1 Introduction

**Multi-modal large language models (MLLMs)** (Song et al., 2023) achieve state-of-the-art understanding and reasoning capabilities across various multi-modal tasks, and are increasingly adopted in fields such as automatic driving (Cui et al., 2024), health care (Liu et al., 2023a), agriculture (Zhu et al., 2024), etc. Capability enhancement and evaluation of MLLMs are highly active areas, with a focus on advancing holistic model capabilities and charting the limits of the MLLMs.

While much of the existing research is concentrated on understanding and reasoning about real-world objects and scenes depicted within images (Fu et al., 2024), other studies have begun to explore models' abilities to interpret and reason over images that convey highly abstract semantic information. As illustrated in Figure 1, these abstractive semantic elements are highly diverse, including charts (Masry et al., 2022), mathematical diagrams (Lu et al., 2024), and more. Such abstractive semantic information is frequently presented at a conceptual level through artificially constructed visual patterns, which are defined by humans and are absent in nature. Effectively reasoning about abstractive image inputs poses an elevated challenge for MLLMs, as it demands not only basic object recognition but also a deeper understanding and interpretation of the complex information encapsulated within these human-defined abstractive visual forms.

Among the diverse array of abstractive images, an important area remains underexplored: **ST**ructured and **A**bstractive **R**easoning (STAR) on images with **M**ulti-**M**odal **R**elational **K**nowledge (MMRK). As illustrated in Figure 1, MMRK consists of multiple multi-modal entities and concepts that are interconnected by abstract relational edges, representing well-organized and structured factual knowledge. Unlike natural or other abstractive images, MMRK offers a flexible and structured format for encoding complex semantic relations, with broad application potential (An et al., 2025). The relational links act as higher-order human-defined abstractions, modeling intricate connections among entities, and thus place greater demands on MLLM's reasoning capabilities. To accurately perform STAR, MLLMs must understand both the entities and the underlying relational structure. However, STAR remains largely unaddressed, with only a few studies (Zhang et al., 2024a; 2025d) briefly investigating this capability, which still face two critical challenges:

(i) **Lack of large-scale data synthesis method for STAR.** From the data perspective, there is a shortage of high-quality MMRK images and corresponding multi-modal instruction data. Automated pipelines for generating diverse and scalable MMRK datasets are missing, along with reliable chain-of-thought (CoT) reasoning annotations needed to improve MLLM's complex thinking and generalization ability.

(ii) **Absence of effective enhancement and evaluation frameworks for STAR.** From a methodology perspective, a systematic training and evaluation framework for STAR is lacking. Existing work (Zhang et al., 2025d) only addresses zero-shot evaluation. Fine-tuning MLLMs on large-scale synthetic data is necessary to effectively enhance their STAR capabilities.

To tackle these challenges, we develop an automatic STAR data synthesis engine that first generates images containing MMRK and then produces instruction data accompanied by fine-grained CoT reasoning. Given the current limitations of MLLMs, our approach leverages multi-modal knowledge graphs (MMKGs) as the data source, which are structured repositories of reliable multi-modal information. We further introduce a variety of MMRK-related tasks during the synthesis process. In addition, we propose a two-stage training framework, combining supervised fine-tuning and preference alignment, to enhance MLLMs' STAR capabilities and introduce a specialized evaluation protocol. Our contribution in this paper can be summarized as follows:

• **Automatic STAR Data Engine.** We introduce the data synthesis engine, which examines MLLM capabilities from a novel perspective called structured and abstractive reasoning (STAR) using MMRK images. Our engine automatically generates high-quality instruction data using large-scale MMKGs with rich relational knowledge, eliminating costly manual annotation. By visualizing sampled multi-modal subgraphs and creating diverse seed STAR tasks, each data instance includes an MMRK image, a task-specific question, and a detailed CoT answer.

• **Comprehensive Training and Evaluation Pipeline.** We propose a systematic pipeline for enhancing and evaluating STAR capabilities in MLLMs. Our two-stage training combines instruction tuning for general competency and preference alignment for targeted optimization, utilizing the data synthesized by our engine. We also establish a dedicated protocol for STAR evaluation.

• **In-depth Experimental Exploration.** We conduct extensive experiment exploration on 5 famous open-source MLLM backbones from 3B to 34B, aiming to identify key factors influencing STAR enhancement. Our results demonstrate that targeted training can substantially improve MLLMs' STAR abilities with abstractive visual information, uncovering the mechanisms that enable accurate reasoning in complex multi-modal semantic contexts. Smaller MLLMs with 3B/7B parameters can outperform mainstream product-level MLLMs like GPT-4o.

## 2 RELATED WORKS

**Multi-modal Large Language Model (MLLM) Enhancement and Evaluation**   MLLMs (Song et al., 2023) extend LLMs with multi-modal understanding and reasoning capabilities by incorporating multi-modal information into LLMs with different connectors (Zhu et al., 2025), which supports diverse modalities such as images (Zhu et al., 2023), audio (Huang et al., 2024), videos (Zhang et al., 2025a). A lot of datasets and benchmarks are developed to enhance and evaluate the specific capabilities of MLLMs, including multi-disciplinary knowledge (Lu et al., 2022; Yue et al., 2024b), fine-grained recognition and perception (Tong et al., 2024), structured chart understanding

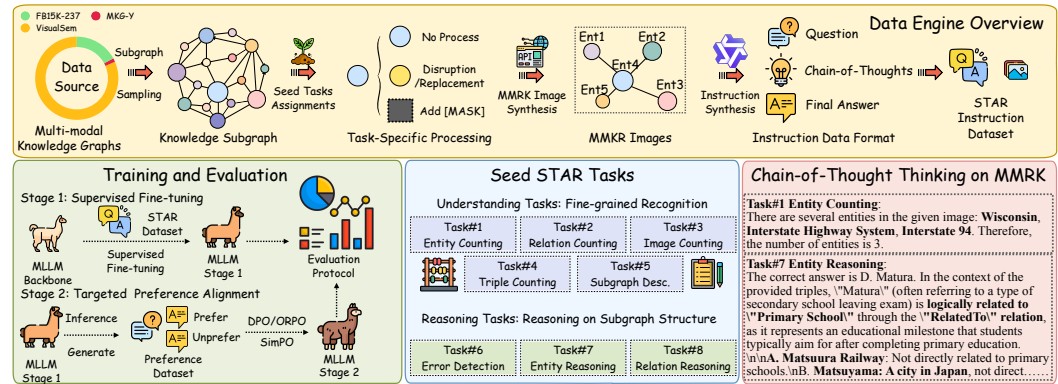

Figure 2: The overview of our data engine, the training pipeline, the seed tasks, and the CoT prompts.

(Masry et al., 2022; Wang et al., 2024), visual mathematical reasoning (Lu et al., 2024; Zhang et al., 2025b), etc. Automatic data synthesis pipelines (Zhang et al., 2024a) are usually designed for these benchmarks to generate high-quality multi-modal instruction data from complex data sources.

**Multi-modal knowledge graphs (MMKGs)** (Chen et al., 2024) consists of structured triple knowledge with multi-modal contents such as entity images (Liu et al., 2019), text descriptions (Yao et al., 2019), and knowledge-grounded audio/videos (Pan et al., 2022). These multi-modal contents enhance traditional triple-based KG with rich semantic information to serve diverse application scenarios by providing multi-modal factual knowledge. In the age of LLM, the combination of LLM and MMKG (Zhang et al., 2024b; Wan & Yu, 2025) attracts widespread attention from both academia and industry, which focuses on leveraging the high-quality multi-modal knowledge to reduce LLM's hallucination (Zhang et al., 2023). This work provides a new perspective to incorporate the abstractive reasoning ability of MLLMs by synthesizing multi-modal instruction data with MMKGs to provide reliable MMRK.

## 3 THE STAR DATA ENGINE

In this section, we introduce our STAR data engine, designed to synthesize images paired with MMRK and corresponding text instructions for constructing high-quality multi-modal instruction datasets tailored to STAR tasks. With this engine, we generate STAR-64K with diverse task types.

### 3.1 DATA ENGINE OVERVIEW

Figure 2 presents the overview of our data engine and synthesis pipeline. With an input subgraph sampled from MMKG with MMRK, the engine generates multi-modal instruction data, comprising pairs of MMRK and text instructions. Note that MMRK is a multi-modal knowledge subgraph, which is the visual modality input for MLLMs. In this work, we define 8 different seed tasks, focusing on MMRK for both structured and abstractive relational reasoning. These seed STAR tasks designed by us are divided into two categories:

**Understanding the MMRK Data.** Before MLLMs can perform complex reasoning, they must accurately identify and describe various components within MMRK. Task types in this category include **Entity Counting** (EC, Task #1), **Relation Counting** (RC, Task #2), **Image Counting** (IC, Task #3), **Triple Counting** (TC, Task #4), and **Subgraph Description** (SD, Task #5). EC, RC, IC, and TC, respectively, require the model to count entities, relations, images, and triples present in MMRK, demonstrating its understanding of fundamental elements. SD asks the MLLM to briefly describe the given visual MMRK, requiring it to grasp the global context of the data. These understanding tasks are inspired by classic visual recognition and understanding benchmarks; however, in the context of MMRK images, recognizing and describing such complex semantic networks in the visual modality becomes significantly more challenging.

**Reasoning on the MMRK Data.** Upon their understanding of MMRK data, MLLMs are expected to integrate information from MMRK with their own knowledge to perform advanced reasoning and predictions, which would be an advanced capability for MLLMs. Therefore, we propose **Error**

**Detection** (ED, Task #6), **Entity Reasoning** (ER, Task #7), and **Relation Reasoning** (RR, Task #8). ED requires MLLMs to detect the anomalous entity that causes a factual error in the given MMRK while ER and RR ask MLLMs to make a choice for a certain missing entity/relation in the given MMRK. The design of these tasks is motivated by classic reasoning tasks like knowledge graph completion (Zhang et al., 2025c) on KGs, and we hope that MLLMs can demonstrate similar reasoning abilities on the visual modality information containing MMRK.

## 3.2 DETAILED SYNTHESIS PIPELINE

Based on the eight seed tasks discussed previously, we devise a five-step pipeline to synthesize MMRK images and text prompts, thereby constructing multi-modal instruction data for STAR tasks.

**Step 1. Data Source.** We select three public MMKGs as our source data: VisualSem (Alberts et al., 2020), FB15K-237 (Liu et al., 2019), and MKG-Y (Xu et al., 2022), which contain million-scale encyclopedic common-sense knowledge with images and entity descriptions as multi-modal contents. The statistical information of the three MMKGs is presented in Appendix A.1. We can denote one MMKG as $\mathcal{KG} = (\mathcal{E}, \mathcal{R}, \mathcal{T}, \mathcal{I}, \mathcal{D})$, where $\mathcal{E}, \mathcal{R}, \mathcal{T}$ represent the entity, relation, and triple sets, respectively. $\mathcal{I}, \mathcal{D}$ are the image and text description sets for entities in the MMKG.

**Step 2. Subgraph Sampling.** Next, we sample knowledge subgraphs $\mathcal{KG}' \subseteq \mathcal{KG}$ where its entity/relation/triple sets are the subsets of the full MMKG. For each sample, an entity is selected as the starting point, followed by a random walk that combines depth-first and breadth-first search until a specified number of entities and relations are collected. To control data complexity, we limit each subgraph to a maximum of 9 entities. After sampling the subgraphs, we also sample images from $\mathcal{I}$ for each entity in $\mathcal{KG}'$ for visualization.

**Step 3. Task-specific Processing.** Subgraph instances are then assigned to each seed task for further task-specific processing. EC, RC, TC, and SD require no additional modification. For IC, we randomly remove some of the images of entities from $\mathcal{I}'$, introducing missing image information and differentiating it from EC. In ED, a single entity is randomly replaced with another from the global entity set to introduce an error. For ER and RR, a target entity or relation is masked in the subgraph by replacing its text and image with a `[MASK]` mark, challenging the model to infer the missing information. After processing, we obtain a modified subgraph $\mathcal{KG}''$ for each subgraph $\mathcal{KG}'$.

**Step 4. MMRK Image Data Synthesis.** We visualize each processed subgraph, converting it into the image modality using a KG visualization tool such as GraphViz (Ellson et al., 2004), formally represented as: $\mathcal{V} \leftarrow \texttt{GraphViz}(\mathcal{KG}'', \mathcal{I}', \mathcal{D}')$ where $\mathcal{I}', \mathcal{D}'$ are the image set and text description set of $\mathcal{KG}'$ respectively. Finally, the MMRK image $\mathcal{V}$ integrates all entities in $\mathcal{KG}''$ with their images and textual descriptions, connected via directional relations. These relational edges provide structured, abstractive information that links entities across multiple modalities, making full understanding and reasoning over such structured abstractions a significant challenge for MLLMs.

**Step 5. Instruction Data Synthesis.** We then synthesize the corresponding instruction data for the synthesized MMRK images. For each $\mathcal{V}$, we prepare an input question $\mathcal{Q}$ and answer $\mathcal{A}$ to form one instruction instance $(\mathcal{V}, \mathcal{Q}, \mathcal{A})$. Distinct question and answer templates are crafted for each seed task type. For a given task, $\mathcal{Q}$ is generated using a fixed template, while the answer is determined by the specific context of $\mathcal{V}$. Meanwhile, the answer $\mathcal{A}$ is divided into two segments: a chain-of-thought (CoT) thinking process and the final answer. Different tasks are associated with specific question and answer templates, as well as distinct methods for generating chain-of-thought (CoT) reasoning. We present the technical details in Appendix A.2 and the instruction templates in Appendix B.

**Information of the Synthetic Data.** With the mentioned pipeline in our data engine, we finally generate 8000 data instances for each task and split them into train/valid/test sets with 8:1:1. Therefore, the full training set consists of 51200 instances, while the validation and test sets consist of 6400 instances of data, respectively. Therefore, we name the 64K data synthesised by us as STAR-64K. Next, we would present the training and evaluation pipeline.

## 4 TRAINING AND EVALUATION PROTOCOL

In this section, we detail our training and evaluation protocol designed to enhance the STAR capability of MLLMs using the large-scale benchmark synthesized by our data engine. Given that

existing MLLMs lack robust STAR abilities, we first employ a two-stage fine-tuning strategy to imbue the models with these capabilities, and then comprehensively assess their performance using our evaluation pipeline.

## 4.1 TWO-STAGE TRAINING FOR CAPABILITY ENHANCEMENT

To strengthen the STAR capability of MLLMs, we propose a two-stage training pipeline. In Stage 1, we perform supervised fine-tuning (SFT) for general capability enhancement; in Stage 2, we apply preference alignment (PA) methods to target specific optimization of failure cases.

**Stage 1: Supervised Fine-tuning for General Enhancement.** We first fine-tune MLLMs with visual instruction data $\mathcal{D}_{sft} = \{(\mathcal{V}_i, \mathcal{Q}_i, \mathcal{A}_i)\}_{i=1}^{N_1}$ synthesised by our data engine. By doing this, MLLMs can learn the basic STAR ability and the basic output format (structured CoT reasoning plus final answer) by fitting on the training data.

**Stage 2: Preference Alignment for Targeted Optimization.** Upon completion of Stage 1, the MLLMs demonstrate baseline competency for simple understanding and reasoning over structured information in visual knowledge graphs. However, we find that a single round of SFT is insufficient to fully unlock the model's potential, especially in complex or error-prone scenarios where hallucinations persist. To address this, we introduce PA methods in Stage 2 for targeted performance improvement on these challenging cases.

The data format for stage 2 can be denoted as $\mathcal{D}_{pa} = \{(\mathcal{V}_i, \mathcal{Q}_i, \mathcal{A}_i^{(p)}, \mathcal{A}_i^{(p)})\}_{i=1}^{N_2}$ where $\mathcal{A}_i^{(p)}, \mathcal{A}_i^{(p)}$ represent a preferred answer and an unpreferred answer for current input $\mathcal{V}_i, \mathcal{Q}_i$. Specifically, we run inference on the training data after Stage 1 and select instances where the model fails to generate correct outputs. For these instances, the gold answers are treated as preferred, and the incorrect, model-generated answers as unpreferred, thus forming the PA dataset $\mathcal{D}_{pa}$. We then adopt PA methods such as DPO (Rafailov et al., 2023) to further optimize the MLLMs, explicitly improving performance on hard cases. By maximizing the likelihood of preferred answers and minimizing that of unpreferred ones, the loss function refines the model's output distribution, thereby boosting accuracy on challenging data. More details of the two-stage pipeline are provided in Appendix A.3.

## 4.2 EVALUATION PROTOCOL

Following the two-stage training, we evaluate MLLM performance on all eight STAR tasks using the protocol below. For all tasks except Task #5, we define ground-truth answers: counting tasks require a numerical value as the answer, while detection/reasoning tasks require a response or selecting a specific entity or relation within the image with MMRK. Accuracy is calculated by comparing predictions against the standard answers.

To assess the quality and correctness of model-generated CoT reasoning, we follow the LLM-as-a-Judge paradigm (Li et al., 2024), leveraging a stronger LLM as an evaluator that scores the generated CoT relative to our gold labels. As Task 5 is open-ended, we assess subgraph descriptions using the same scoring approach as for CoT evaluation. This comprehensive protocol enables holistic assessment of MLLM STAR capabilities using a diverse set of metrics.

## 5 EXPERIMENTS AND ANALYSIS

In this section, we introduce the detailed experiment settings and present our experiment results and further analysis, focusing on the following **research questions (RQ)**:

- **RQ1**: Does the two-stage training pipeline enhance the STAR capabilities of MLLM?
- **RQ2**: Can the different tasks we design influence each other with positive or negative transfer?
- **RQ3**: What scale of data is required to incorporate STAR capabilities?
- **RQ4**: Does the CoT process have a positive effect on the final performance?
- **RQ5**: How much do the entity images and texts influence the final model performance?
- **RQ6**: Are there any intuitive cases to show the performance of MLLMs after two-stage training?

Table 1: The main experiment results on two-stage training on 5 open-source MLLMs. For stage 1(S1), we conduct two groups of experiments S1(single) and S1(Full), representing SFT on single task/full data. For stage 2(S2), we employ three classic PA methods including DPO/OPRO/SimPO.

| Experiment Settings | | | Task#1 | | Task#2 | | Task#3 | | Task#4 | | Task#5 | Task#6 | | Task#7 | | Task#8 | | AVG |
|---|---|---|---|---|---|---|---|---|---|---|---|---|---|---|---|---|---|---|
| | | | ACC | CoT | ACC | CoT | ACC | CoT | ACC | CoT | Score | ACC | CoT | ACC | CoT | ACC | CoT | |
| **QVQ-72B** | | | 30.75 | - | 8.25 | - | 5.50 | - | 42.50 | - | 50.13 | 4.63 | - | 25.38 | - | 16.00 | - | 22.89 |
| **Qwen2.5-72B** | | | 38.25 | - | 65.38 | - | 8.63 | - | 39.13 | - | 65.00 | 0.25 | - | 40.38 | - | 52.88 | - | 38.74 |
| **GPT-4v** | | | 37.75 | - | 41.25 | - | 14.00 | - | 40.00 | - | 59.25 | 3.63 | - | 29.83 | - | 39.13 | - | 33.11 |
| **GPT-4o-mini** | | | 67.50 | - | 72.25 | - | 29.88 | - | 31.25 | - | 69.13 | 3.50 | - | 29.25 | - | 23.00 | - | 40.72 |
| **GPT-4o** | | | 43.75 | - | 56.33 | - | 17.38 | - | 34.50 | - | 82.38 | 2.73 | - | 53.88 | - | 40.00 | - | 41.37 |
| Qwen2.5-VL | 3B | Zero-shot | 18.25 | - | 20.13 | - | 3.50 | - | 12.75 | - | 57.71 | 6.25 | - | 47.63 | - | 38.25 | - | 25.56 |
| | | S1(Single) | 51.00 | 62.07 | 56.63 | 74.75 | 10.38 | 28.38 | 20.13 | 31.90 | 58.31 | 20.37 | 32.34 | 52.75 | 53.76 | 64.50 | 54.00 | 41.76 |
| | | S1(Full) | 42.75 | 52.67 | 67.00 | 79.74 | 57.13 | 29.17 | 23.50 | 31.87 | 59.94 | 37.25 | 32.00 | 61.13 | 55.88 | 77.25 | 56.00 | 53.24 |
| | | S2(DPO) | 55.50 | 73.28 | 89.25 | 95.67 | 66.88 | 65.37 | 26.13 | 51.77 | 66.64 | 37.50 | 38.42 | 60.00 | 60.94 | 76.85 | 64.77 | 59.84 |
| | | S2(ORPO) | 39.00 | 64.79 | 84.62 | 94.06 | 59.00 | 60.29 | 17.88 | 43.81 | 66.81 | 37.75 | 39.00 | 59.63 | 61.21 | 77.63 | 65.79 | 55.29 |
| | | S2(SimPO) | 71.00 | 79.03 | 89.38 | 96.82 | 37.25 | 52.31 | 28.13 | 53.23 | 67.92 | 37.62 | 40.06 | 59.13 | 60.89 | 78.25 | 66.78 | 58.59 |
| | 7B | Zero-shot | 6.13 | - | 12.25 | - | 0.13 | - | 13.13 | - | 68.62 | 0.75 | - | 26.00 | - | 42.88 | - | 21.24 |
| | | S1(Single) | 77.13 | 82.47 | 91.13 | 95.85 | 65.88 | 71.10 | 24.75 | 54.96 | 74.85 | 52.75 | 42.93 | 64.38 | 65.84 | 77.63 | 68.24 | 66.06 |
| | | S1(Full) | 64.88 | 81.79 | 92.75 | 97.38 | 71.37 | 76.70 | 27.62 | 54.07 | 75.71 | 55.87 | 45.23 | 67.50 | 69.40 | 80.13 | 71.52 | 66.98 |
| | | S2(DPO) | 66.50 | 82.11 | 94.00 | 97.79 | 73.50 | 77.32 | 30.25 | 58.67 | 76.44 | 58.63 | 46.10 | 69.37 | 68.79 | 82.00 | 72.35 | 68.84 |
| | | S2(ORPO) | 65.75 | 81.96 | 93.38 | 98.89 | 71.88 | 77.09 | 27.38 | 54.45 | 76.65 | 56.38 | 47.11 | 70.00 | 68.96 | 79.75 | 71.20 | 67.65 |
| | | S2(SimPO) | 69.63 | 82.96 | 93.75 | 97.76 | 75.38 | 78.55 | 29.00 | 56.77 | 76.32 | 57.75 | 46.52 | 68.50 | 68.13 | 81.50 | 71.95 | 68.98 |
| | 32B | Zero-shot | 55.50 | - | 70.25 | - | 14.88 | - | 1.63 | - | 72.08 | 5.38 | - | 37.50 | - | 40.38 | - | 37.20 |
| | | S1(Single) | 77.25 | 81.29 | 88.00 | 83.07 | 57.75 | 64.35 | 23.13 | 46.87 | 69.98 | 41.50 | 41.05 | 64.63 | 65.11 | 77.00 | 65.24 | 62.41 |
| | | S1(Full) | 67.75 | 79.84 | 93.63 | 99.70 | 63.13 | 70.21 | 27.50 | 53.93 | 75.07 | 54.00 | 44.16 | 73.50 | 68.90 | 81.75 | 70.59 | 67.04 |
| | | S2(DPO) | 73.25 | 82.12 | 94.50 | 97.73 | 65.75 | 72.46 | 32.50 | 62.61 | 72.98 | 52.38 | 44.67 | 71.25 | 69.95 | 81.63 | 70.75 | 68.03 |
| | | S2(ORPO) | 60.38 | 78.65 | 93.75 | 97.39 | 60.75 | 69.98 | 25.25 | 55.44 | 73.36 | 52.75 | 46.07 | 69.63 | 68.92 | 80.62 | 69.35 | 64.56 |
| | | S2(SimPO) | 76.37 | 82.01 | 89.25 | 92.25 | 66.63 | 73.23 | 32.75 | 60.29 | 74.16 | 54.38 | 45.44 | 71.38 | 69.78 | 82.25 | 71.59 | 68.40 |
| LLaVA-1.5/NEXT | 7B | Zero-shot | 11.75 | - | 2.13 | - | 12.88 | - | 4.38 | - | 20.27 | 1.13 | - | 36.50 | - | 59.88 | - | 18.62 |
| | | S1(Single) | 34.13 | 28.91 | 43.25 | 66.18 | 23.38 | 28.99 | 11.25 | 22.27 | 25.77 | 2.25 | 18.68 | 33.63 | 37.75 | 61.35 | 41.32 | 29.38 |
| | | S1(Full) | 70.38 | 39.02 | 66.75 | 83.89 | 60.75 | 34.50 | 22.25 | 27.30 | 33.24 | 6.50 | 22.93 | 54.50 | 41.76 | 79.62 | 48.61 | 49.25 |
| | | S2(DPO) | 71.25 | 49.41 | 86.13 | 92.02 | 62.63 | 46.40 | 44.75 | 41.84 | 42.34 | 19.37 | 29.23 | 60.63 | 59.50 | 80.25 | 58.59 | 58.42 |
| | | S2(ORPO) | 71.25 | 49.72 | 86.13 | 92.27 | 63.25 | 46.01 | 45.50 | 42.22 | 42.87 | 19.75 | 28.99 | 60.00 | 59.45 | 79.88 | 59.78 | 58.58 |
| | | S2(SimPO) | 67.38 | 48.86 | 86.13 | 92.07 | 63.50 | 46.75 | 39.00 | 41.69 | 42.41 | 20.13 | 28.97 | 59.50 | 57.65 | 80.13 | 58.46 | 57.27 |
| | 34B | Zero-shot | 12.25 | - | 7.63 | - | 8.25 | - | 19.63 | - | 59.31 | 0.63 | - | 45.88 | - | 54.88 | - | 26.06 |
| | | S1(Single) | 57.13 | 75.12 | 81.50 | 90.20 | 68.38 | 52.77 | 84.00 | 40.78 | 65.23 | 58.50 | 31.33 | 52.38 | 56.75 | 67.63 | 58.91 | 59.03 |
| | | S1(Full) | 97.50 | 86.05 | 96.75 | 98.04 | 85.00 | 80.79 | 68.13 | 64.35 | 72.02 | 66.63 | 38.67 | 75.25 | 60.90 | 85.00 | 66.39 | 80.79 |
| | | S2(DPO) | 97.88 | 93.05 | 99.25 | 99.62 | 98.88 | 89.98 | 66.87 | 83.30 | 79.43 | 66.38 | 46.94 | 74.75 | 72.32 | 84.62 | 75.79 | 83.51 |
| | | S2(ORPO) | 97.88 | 92.47 | 99.25 | 99.43 | 98.75 | 89.71 | 70.13 | 82.89 | 79.86 | 66.87 | 46.98 | 76.00 | 73.00 | 85.25 | 75.55 | 84.25 |
| | | S2(SimPO) | 98.25 | 92.96 | 99.25 | 99.29 | 98.75 | 90.09 | 62.25 | 82.31 | 79.42 | 67.00 | 45.46 | 73.88 | 71.86 | 85.87 | 76.20 | 83.08 |

## 5.1 EXPERIMENT SETTINGS

**Baselines.** In our experiments, we utilize Qwen2.5-VL-3B/7B/32B (Bai et al., 2025), LLaVA-1.5-7B (Liu et al., 2023b), and LLaVA-NEXT-34B (Liu et al., 2024) as MLLM backbones. For each backbone, we report three groups of results: (1) zero-shot performance without any finetuning, (2) results after stage 1 training (Vanilla SFT), and (3) results after both stage 1 and stage 2 training (SFT + PA). In the SFT stage, we experiment with two settings: training on single-task data and on the full STAR-64K dataset. For PA, we employ three mainstream PA methods: DPO (Rafailov et al., 2023), ORPO (Hong et al., 2024), and SimPO (Meng et al., 2024). We include the zero-shot results of larger models, including QVQ-72B (Team, 2024), Qwen2.5-VL-72B (Bai et al., 2025), GPT-4V (OpenAI, 2023), GPT-4o-mini, and GPT-4o (OpenAI, 2024), for comprehensive comparison.

**Hyper-parameter Settings** We implement our two-stage training (SFT&PA) and inference process with two famous open-source projects: LLaMA-Factory (Zheng et al., 2024) and vLLM (Kwon et al., 2023). We conduct our experiments on $8\times$ NVIDIA A100 GPUs. The max sequence length is set to 8192 and the global batch size to 8 with BF16 precision. We train MLLMs with LoRA (Hu et al., 2022) and search the rank in $\{8, 16\}$. AdamW (Loshchilov & Hutter, 2019) optimizer is used for both training stages with a cosine scheduler. For stage 1, we set the training epoch to 3. The learning rate is searched in $\{1e^{-5}, 1e^{-4}, 3e^{-4}\}$. For stage 2, we train MLLMs with further 1 epoch on the checkpoints of stage 1 and set the learning rate to $1e^{-6}$. The PA data scale $N_2 = 16663$ according to the bad cases we collect from the training set of MMRK-64K after stage 1.

For evaluation, we employ Qwen2.5-VL-72B (Bai et al., 2025) as an LLM judger to score model predictions (CoT and unstructured zero-shot results) against the golden labels, providing a more objective and scalable assessment. The evaluation prompt templates used are detailed in Appendix B.

## 5.2 MAIN EXPERIMENT RESULTS (RQ1)

We summarize the main experimental results in Table 1, which reports the performance of five MLLM backbones on STAR tasks before and after the proposed two-stage training pipeline. For

each backbone, we evaluate six experimental settings: (1) zero-shot performance, (2–3) stage 1 (SFT) trained with either single-task data or the full STAR-64K dataset, and (4–6) stage 2 (PA) using three alternative methods, DPO, ORPO, and SimPO. Based on these results, we draw the following key observations:

**Existing mainstream MLLMs fail on the STAR tasks.** From the zero-shot results of OpenAI's GPT models and other open-source MLLMs, it is evident that current leading MLLMs struggle with STAR tasks, indicating that their generalization capabilities do not readily extend to synthetic images and MMKR scenarios. Notably, after SFT with single-task data, Qwen2.5-VL-3B achieves a comparable or even slightly better overall accuracy than much larger models such as Qwen2.5-VL-72B and GPT-4o (41.76% vs. 38.74% / 41.37%). This suggests that the limited performance of current MLLMs on STAR tasks is mainly due to insufficient relevant data during their training phases. Simply applying SFT on single-task data can partially unlock this latent capability, but still requires further fine-grained optimization.

**Two-stage pipeline progressively improves the STAR capabilities.** Comparing results across the full STAR-64K dataset, we observe that stage 1 SFT leads to substantial improvements over zero-shot performance as models adapt to synthetic multimodal instructions and learn to solve diverse task types. Stage 2 (PA) delivers additional performance gains, albeit smaller than those achieved in stage 1, highlighting the complementary effect of preference alignment. The three PA methods exhibit strong generality and consistently improve results across different backbones. In terms of backbone comparison, LLaVA-1.5-7B consistently underperforms relative to Qwen2.5-VL-7B, whereas LLaVA-NEXT-34B demonstrates clear superiority over Qwen2.5-VL-32B, particularly in counting-related tasks such as EC, RC, IC, and TC. This suggests that both architecture design and scale, alongside our training paradigms, are crucial for advancing STAR performance.

## 5.3 TRANSFERABILITY EXPERIMENTS (RQ2)

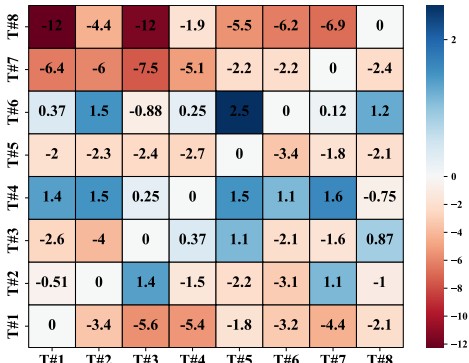

Figure 3: The task-wise transferability experiments. The y-axis is the basic task.

In addition to the main experiments, we conduct a series of supplementary SFT trials using single-task datasets, aiming to investigate the transferability among different STAR tasks. Compared to joint training on the full dataset, single-task training generally results in diminished performance across most tasks, with Task #1 being a notable exception. This suggests that mixed training with diverse multi-task instructions promotes knowledge transfer across tasks and collectively enhances overall model performance. The unique case of Task #1, which relies predominantly on basic entity recognition abilities, indicates that this fundamental capability is not further improved by subsequent training on more complex recognition or reasoning tasks. Complex tasks, on the other hand, present greater learning challenges for MLLMs, while simpler recognition tasks enable models to better capture underlying patterns in MMKR images. Additionally, to further probe task transferability, we conduct supplementary SFT experiments by pairing tasks during SFT. As illustrated in Figure 3, pairwise task combinations reveal more nuanced mutual enhancement effects, with Tasks 4 and 6 benefiting especially from being trained alongside other tasks. This observation is consistent with findings from the main experiments. Overall, these results suggest that MLLMs can develop emergent STAR capabilities through training on a broader and more complex set of tasks, gradually generalizing to new or related tasks. However, the emergence and effectiveness of such generalization critically depend on the diversity and richness of the training data provided.

## 5.4 SCALABILITY EXPERIMENTS (RQ3)

We further investigate the scalability of the STAR data, aiming to determine the data volume required to instill fundamental STAR capabilities in MLLMs. In Figure 4, we present the answer and CoT

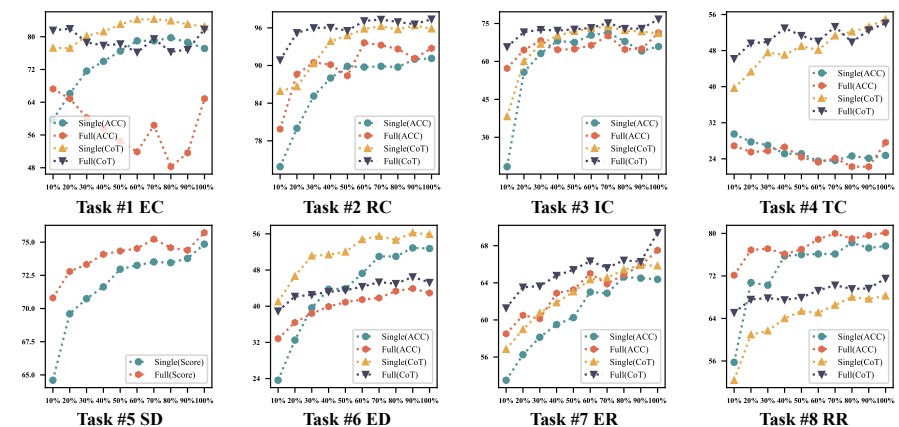

Figure 4: The scalability experiments Qwen2.5-VL-7B for 8 STAR tasks.

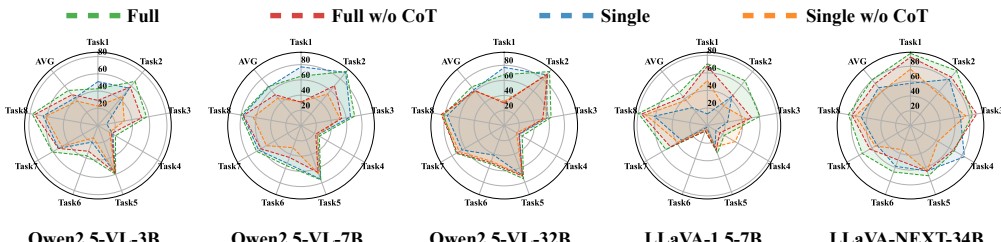

Figure 5: Ablation study on the effectiveness of CoT prompts in the instruction data.

quality results of 8 STAR tasks trained under single-task and full-data settings from 10% to 100% data. The experimental result indicates that, for all other tasks except for Task #1 and Task #4, most tasks exhibit a clear trend of increasing STAR capability as the training data scale grows, consistent with established scale laws in data-driven learning.

For Task #1, ACC fluctuates with increasing dataset size, whereas CoT quality steadily improves, indicating that while the model's overall counting accuracy does not consistently increase, its precision in entity recognition within CoT becomes progressively better. This can be attributed to the fact that MLLMs possess an inherent counting ability that does not markedly improve with further training. In contrast, their aptitude for recognizing and distinguishing objects in MMKR images advances noticeably. Task #7 follows a similar pattern: ACC remains mostly unchanged, yet CoT quality continues to climb, suggesting that the model is refining its content identification in MMKR images even as its aggregate counting capability remains limited by architectural constraints. Considering the main experiments, the upper limit of this capability on the Qwen2.5-VL-7B model aligns closely with these results. To surpass current limitations, it is necessary to utilize more powerful backbones, as further scaling of training data alone yields diminishing returns for certain task types.

## 5.5 ABLATION STUDY (RQ4&RQ5)

To further investigate the key factors contributing to the STAR performance, we conduct two ablation studies for the CoT prompts and the modality information incorporated in the MMKR images.

**The effectiveness of CoT.** As mentioned before, we construct CoT prompts for different tasks to guide MLLMs in identifying and reasoning over the relevant elements within the given MMRK image. To further explore its effectiveness, we conduct several experiments that remove the CoT prompts in the training data. As shown in Figure 5, the STAR performance of all 5 different MLLMs consistently degrades when the CoT prompts are omitted, regardless of whether models are trained on single-task or full multi-task data. These results demonstrate that CoT contributes to performance improvement, providing effective guidance for models to think and solve STAR tasks.

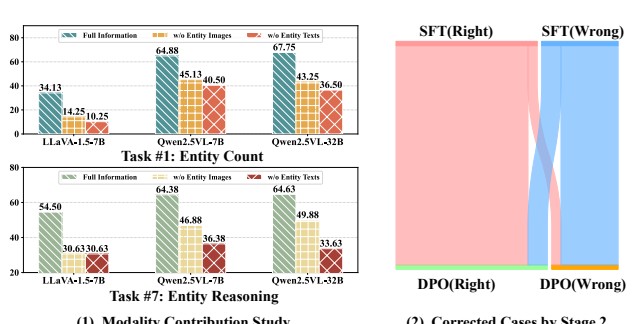

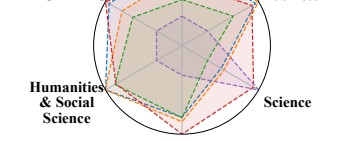

(1). Modality Contribution Study       (2). Corrected Cases by Stage 2       (3). Commonsense Knowledge Retention

Figure 6: The modality contribution experiments and case studies.

Table 2: Study of modality contribution on full datasets.

| Setting | | Task1 | Task2 | Task3 | Task4 | Task5 | Task6 | Task7 | Task8 |
|---|---|---|---|---|---|---|---|---|---|
| Qwen2.5-VL 7B | w/o ent. images | 55.50 | 75.88 | 48.62 | 26.63 | 67.99 | 32.00 | 52.63 | 65.75 |
| | w/o ent. texts | 59.13 | 74.62 | 47.88 | 25.37 | 67.90 | 34.87 | 41.50 | 68.12 |
| | full dataset | 64.88 | 92.75 | 71.37 | 27.62 | 75.71 | 55.87 | 67.50 | 80.13 |
| Qwen2.5-VL 32B | w/o ent. images | 49.75 | 83.25 | 42.25 | 29.88 | 66.05 | 29.63 | 42.50 | 68.00 |
| | w/o ent. texts | 58.25 | 82.25 | 41.00 | 25.88 | 65.61 | 28.63 | 46.25 | 66.88 |
| | full dataset | 67.75 | 93.63 | 63.13 | 27.50 | 75.07 | 54.00 | 73.50 | 81.75 |
| LLaVA-1.5 7B | w/o ent. images | 33.87 | 68.13 | 38.38 | 20.50 | 33.20 | 6.00 | 31.50 | 70.13 |
| | w/o ent. texts | 37.13 | 67.50 | 42.13 | 21.25 | 33.09 | 6.50 | 33.38 | 69.62 |
| | full dataset | 70.38 | 66.75 | 60.75 | 22.25 | 33.24 | 6.50 | 54.50 | 79.62 |

**Modality contribution.** To synthesize the MMRK images, we incorporate the entity images and texts in the MMKG to construct semantic-rich visualized subgraphs. To assess their impact on them, we conduct SFT experiments for Task #1 and Task #7 by re-synthesizing MMKR images without entity images or without texts. These two tasks are entity-centric and are greatly affected by the completeness of entity information. As shown in Figure 6(1), performance drops noticeably on both tasks when either modality is removed, underscoring the value of both visual and textual entity information for effective MLLM reasoning. Notably, omitting entity texts leads to a greater decline, suggesting that textual information is particularly critical. Besides, we present the full task results in Figure 2, which highlight the importance of multi-modal entity information, with textual content playing a dominant role in the STAR tasks.

## 5.6 Case Study (RQ6)

To provide a more intuitive understanding of the effectiveness of our two-stage training pipeline, we present a case study in this section. Before SFT (stage 1), the model's STAR performance is notably poor, but undergoes marked improvement following SFT. The main experiments further reveal that the second-stage PA process delivers additional gains in STAR accuracy. To identify where these improvements occur, we analyze the distribution of the model's inference results after SFT and subsequent DPO training, as shown in Figure 6(2). Our analysis shows that PA training in the second stage corrects a substantial proportion of erroneous outputs generated after SFT, although some errors persist in a small number of cases. Overall, the higher rate of corrected test predictions confirms a net improvement in model performance. Moreover, expanded case studies presented in Appendix C demonstrate that the stage 2 PA not only increases answer accuracy, but also significantly reduces hallucinations in the CoT reasoning process. Meanwhile, as illustrated in Figure 6(3), we assess the retention of Qwen2.5-VL-7B's commonsense knowledge at various stages using MMMU (Yue et al., 2024a). The results show that two-stage training with STAR-64K not only preserves but, in some domains—such as arts and business—even enhances commonsense knowledge. This demonstrates that STAR capabilities can be effectively integrated into existing MLLMs, improving their performance while maintaining their commonsense reasoning abilities, which would be a win-win strategy. Further details of the commonsense knowledge retention are in Appendix C.

## 6 Conclusion

In this paper, we investigate structured and abstractive reasoning on images enriched with multi-modal relational knowledge for MLLMs. To address this research gap, we design a data engine that synthesizes STAR instruction data and introduces STAR capabilities to models through a customized training and evaluation pipeline. We systematically assess model performance and thoroughly validate the extent of current MLLMs' STAR capabilities, as well as the improvements enabled by our pipeline. Furthermore, we conduct comprehensive analyses of task transferability, data scalability, and design reasonability with intuitive cases to show the effectiveness of our design.

## ETHICS STATEMENT

In this paper, we utilize three open-source knowledge graphs (KGs) as our data sources, which we then modify to generate new datasets. Additionally, the primary MLLM backbones we employ are mainstream open-source models. We did not collect data or conduct computational experiments in ways that violated scientific ethics. Therefore, our work does not involve any ethical issues.

## REPRODUCIBILITY STATEMENT

We detail the entire pipeline in our methodology section and elaborate on the hyperparameters involved in the experimental settings. Additionally, we provide the relevant pipeline code in the supplementary materials to ensure the reproducibility of this work.

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

Table 3: Statistical information about the MMKG data source used in our data engine.

| Dataset | Entity | Relation | Triple | Data Source |
|---------|--------|----------|--------|-------------|
| **FB15K-237** | 14541 | 237 | 310116 | FreeBase |
| **MKG-Y** | 15000 | 16 | 26638 | YAGO |
| **VisualSem** | 89896 | 13 | 1481007 | Wikipedia, ImageNet, BabelNet |

# A  DETAILS OF THE DATA ENGINE AND TRAINING PIPELINE

## A.1  DETAILS OF OUR DATA SOURCE

We present the detailed information of the MMKGs used in our data engine in Table 3, which includes FB15K-237 (Bollacker et al., 2008), MKG-Y (Xu et al., 2022), and VisualSem (Alberts et al., 2020). They are constructed from heterogeneous knowledge bases like FreeBase (Bollacker et al., 2008), YAGO (Suchanek et al., 2007), WikiPedia (Vrandecic & Krötzsch, 2014), ImageNet (Deng et al., 2009), and BabelNet (Navigli & Ponzetto, 2010). These MMKGs encompass diverse entities and relation types, with the knowledge triples they form containing a wide range of encyclopedic and commonsense knowledge. We utilized these MMKGs to construct a large-scale STAR dataset based on our data engine.

## A.2  DETAILS OF INSTRUCTION SYNTHESIS

For EC/RC/IC/TC, their CoT prompts are based on several CoT templates that guide MLLMs to recognize the detailed information in $\mathcal{V}$, and the final answer is the proper number of elements.

For SD, its CoT and final answer are combined. We generate a paragraph of words to describe the current MMRK by prompting a strong LLM with the detailed texts of the subgraph.

For ED, the final answer is the disrupted entity, and we generate a CoT for error analysis based on the given knowledge contexts. For ER and RR, the final answer is the option where the entity/relation is masked, and we generate CoT to guide MLLMs' thinking and reasoning.

The overarching principle for constructing CoT is to utilize information from the subgraph before visualization as prompts for the LLM, thereby guiding the LLM to generate the corresponding reasoning process. Compared to presenting MLLMs with a synthesized MMRK image, this approach

yields higher-quality CoT data with fewer hallucinations. Current MLLMs lack sufficient understanding of MMRK images, leading to numerous errors. However, when provided with accurate text prompts of subgraphs, LLMs can generate appropriate results.

## A.3 DETAILS OF TRAINING

**Stage 1. Supervised Fine-tuning**    The SFT process, following the next token prediction paradigm, can be denoted as:

$$\mathcal{L}_{sft} = -\mathbb{E}_{(\mathcal{V}_i, \mathcal{Q}_i, \mathcal{A}_i) \sim \mathcal{D}_{sft}} \left[ \log P_{\mathcal{M}}(\mathcal{A}_i \mid \mathcal{V}_i, \mathcal{Q}_i) \right] \tag{1}$$

where $P_{\mathcal{M}}$ represents the conditional probability of the current answer given by the MLLM $\mathcal{M}$.

**Stage 2. Preference Alignment**    The PA process with DPO can be denoted as:

$$\mathcal{L}_{pa} = -\mathbb{E} \left[ \log \sigma \beta \left( \log \frac{P_{\mathcal{M}}(\mathcal{A}_i^{(p)} \mid \mathcal{V}_i, \mathcal{Q}_i)}{P_{\mathcal{M}_{ref}}(\mathcal{A}_i^{(p)} \mid \mathcal{V}_i, \mathcal{Q}_i)} - \log \frac{P_{\mathcal{M}}(\mathcal{A}_i^{(u)} \mid \mathcal{V}_i, \mathcal{Q}_i)}{P_{\mathcal{M}_{ref}}(\mathcal{A}_i^{(u)} \mid \mathcal{V}_i, \mathcal{Q}_i)} \right) \right] \tag{2}$$

where $\sigma$ is the sigmoid function and $\beta$ is the temperature hyper-parameter. $\mathcal{M}_{ref}$ represents the reference model, which is the MLLM trained after stage 1 in practice. Note that in our experiments, other improved versions of DPO, such as ORPO (Hong et al., 2024) and SimPO (Meng et al., 2024), are also employed in stage 2.

## B  INSTRUCTION TEMPLATES

This section presents the instruction templates used in our MMRK data synthesis and performance evaluation process. The instruction templates we presented in this section include: Figure 7: the question templates for 8 STAR tasks; Figure 8: the answer templates (w/ CoT) for 8 STAR tasks; Figure 9: the instruction template used for subgraph description (Task #5) quality evaluation; Figure 10: the instruction template used for CoT quality evaluation for other tasks. Qwen2.5-72B.

## C  CASE STUDY

In this section, we present more detailed case studies to illustrate the effectiveness of the two-stage training pipeline. We present three cases in Figure 11, 12, 13. From these cases, we observe a common pattern: both pure zero-shot results and those trained solely on S1 exhibit severe hallucinations. MLLM generates numerous entities and relations in CoT that are entirely unrelated to the MMRK within the current image, leading to erroneous final answers. However, through targeted optimization in S2, MLLM's hallucinations are suppressed, and its accuracy is evidenced by statistical results. This indicates that our two-stage design has indeed functioned as we expected.

For the commonsense knowledge retention experiments, we employ MMMU (Yue et al., 2024a), which is one of the most popular MLLM benchmarks for commonsense knowledge evaluation. MMMU consists of diverse subjects which can be categorized into arts & designs, business, science, health & medical, humanities & social science, and tech & engineering. We evaluated Qwen2.5-VL-7B's performance across these six domains on its validation set before and after two-stage training. To enable clearer comparison, we applied max-min normalization to the results. The findings reveal that MLLM models trained through the second phase of the STAR task demonstrate improved performance across all domains except science. This contrasts with the common observation that models lose generalizability after instruction-based fine-tuning. This demonstrates that training on the STAR task can activate or enhance the common-sense knowledge of MLLMs to a certain extent without causing catastrophic forgetting. Consequently, it can be integrated as a new capability into existing MLLMs, which underscores the significance of our research.

## D  THE USE OF LARGE LANGUAGE MODELS

The primary research subject of this paper is LLM & MLLM. Additionally, LLMs are employed **as a general assistant** for code debugging and polishing certain paragraphs. Core idea conception, experimental design, and paper writing are completed by human authors.

---

**Question Templates for 8 STAR Tasks**

**Task #1: Entity Counting**
<image>Given the multi-modal knowledge graph. Please count the number of entities in it.
**Task #2: Relation Counting**
<image>Given the multi-modal knowledge graph. Please count the number of different relations in it.
**Task #3: Image Counting**
<image>Given the multi-modal knowledge graph. Please count the number of entities that have image information in the given knowledge graph.
**Task #4: Triple Counting**
<image>Given the multi-modal knowledge graph. Please count the number of knowledge triples in it.
**Task #5: Subgraph Description**
<image>Given the multi-modal knowledge graph. Please describe the knowledge presented by it.
**Task #6: Error Detection**
<image>Given the multi-modal knowledge graph. Please point out the wrong entity in it.
**Task #7: Entity Reasoning**
<image>Given the multi-modal knowledge graph. One entity in it is replaced by [MASK]. Please select one correct entity from the options.
**Task #8: Relation Reasoning**
<image>Given the multi-modal knowledge graph. One relation in it is replaced by [MASK]. Please select one correct relation from the options.

Figure 7: The question templates for STAR tasks.

---

**Answer (w/ CoT) Templates for 8 STAR Tasks**

**Task #1: Entity Counting**
<think> There are several entities in the given multi-modal knowledge graph: {ENT1, ENT2, ......, ENT K} Therefore, the number of entities is {ENTITY NUMBER} </think> <answer>{ENTITY NUMBER}</answer>
**Task #2: Relation Counting**
<think> There are several different relations in the given multi-modal knowledge graph: {REL1, REL2, ......, REL K} Therefore, the number of different relations is {RELATION NUMBER} </think> <answer>{RELATION NUMBER}</answer>
**Task #3: Image Counting**
<think> There are several entities with images in the given multi-modal knowledge graph: {ENT1, ENT2, ......, ENT M} Other entities without images are: {ENT1, ENT2, ......, ENT N} Therefore, the number of entities is {IMAGE NUMBER} </think> <answer>{IMAGE NUMBER}</answer>
**Task #4: Triple Counting**
<think> There are several knowledge triples in the given multi-modal knowledge graph: Therefore, the number of triples is </think> <answer></answer>
**Task #5: Subgraph Description**
Description of the subgraph.
**Task #6: Error Detection & Task #7: Entity Reasoning & Task #8: Relation Reasoning**
<think> CoT annotated by LLM </think> <answer>{OPTION}</answer>

Figure 8: The answer templates for STAR tasks.

Instruction Template for Task5 Quality Evaluation

As an automated answer-scoring system, please evaluate the similarity between the model's generated responses and the correct answers.
Both of the standard answer and model generated answer are describing a knowledge graph with several sentences.
You must determine whether the key entities, relations, and knowledge mentioned in the model's generated response align with the standard answer.
Ultimately, output an integer between 0 and 100, where a higher number indicates greater similarity. Below are our defined basic scoring rules:
- 0 points: No similarity at all
- 1 to 40 points: Minor information overlap
- 40 to 60 points: Moderate information overlap
- 60 to 90 points: Substantial and detailed information overlap
- Above 90 points: Virtually identical, with only minor syntactic variations
Standard Answer:
Model Generated Answer:
Please response a number for the score directly. Do not provide any other text in the response.

Figure 9: The instruction template used for subgraph description (Task #5) quality evaluation with Qwen2.5-72B.

Instruction Template for Chain-of-thought Quality Evaluation

As an automated answer-scoring system, please evaluate the similarity between the model's generated thought process and the golden label for thought process.
You must determine whether the key entities, relations, and knowledge mentioned in the model's generated thought process align with the standard answer.
Ultimately, output an integer between 0 and 100, where a higher number indicates greater similarity. Below are our defined basic scoring rules:
- 0 points: No similarity at all
- 1 to 30 points: Minor information overlap
- 30 to 60 points: Moderate information overlap
- 60 to 90 points: Substantial and detailed information overlap
- Above 90 points: Virtually identical, with only minor syntactic variations
- If both the thoght process and the final answer match the golden label, full score is awarded.
- If the reasoning process is incorrect but the final answer is correct, partial score may be given.
- If neither the reasoning process nor the final answer is correct, a lower score is assigned.
**Standard Thought Process**: {}
**Model Generated Thought Process**: {}
Please response a number for the score directly. Do not provide any other text in the response.

Figure 10: The instruction template used for CoT quality evaluation with Qwen2.5-72B.

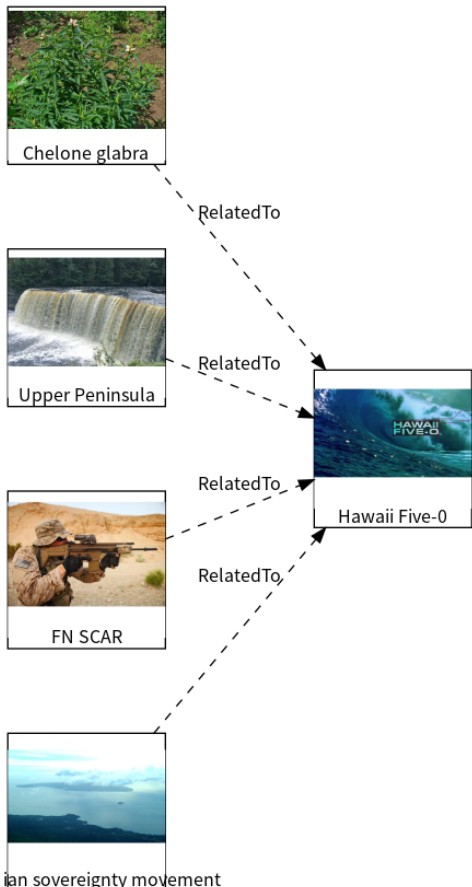

Figure 11: The MMKR image case for Task #1.

---

**Case for Task #1**

**Zero-shot Answer:** The multi-modal knowledge graph contains 10 entities.

**Stage 1 Answer:**

\n<think>\nThere are several entities in the given multi-modal knowledge graph:\nUpper Peninsula\nChelone glabra\nChelone glabra\nHawaii Five-0\nFN SCAR\nIndigenous sovereignty movement\nHawaii\nTherefore, the number of entities is 7\n</think>\n<answer>7</answer>\n

**Stage 2(DPO) Answer:**

\n<think>\nThere are several entities in the given multi-modal knowledge graph:\nHawaii Five-0\nChelone glabra\nUpper Peninsula\nFN SCAR\nIndigenous sovereignty movement\nTherefore, the number of entities is 5\n</think>\n<answer>5</answer>\n

**Golden Answer:**

\n<think>\nThere are several entities in the given multi-modal knowledge graph:\nChelone glabra\nHawaii Five-0\nUpper Peninsula\nFN SCAR\nHawaiian sovereignty movement\nTherefore, the number of entities is 5\n</think>\n<answer>5</answer>\n\n

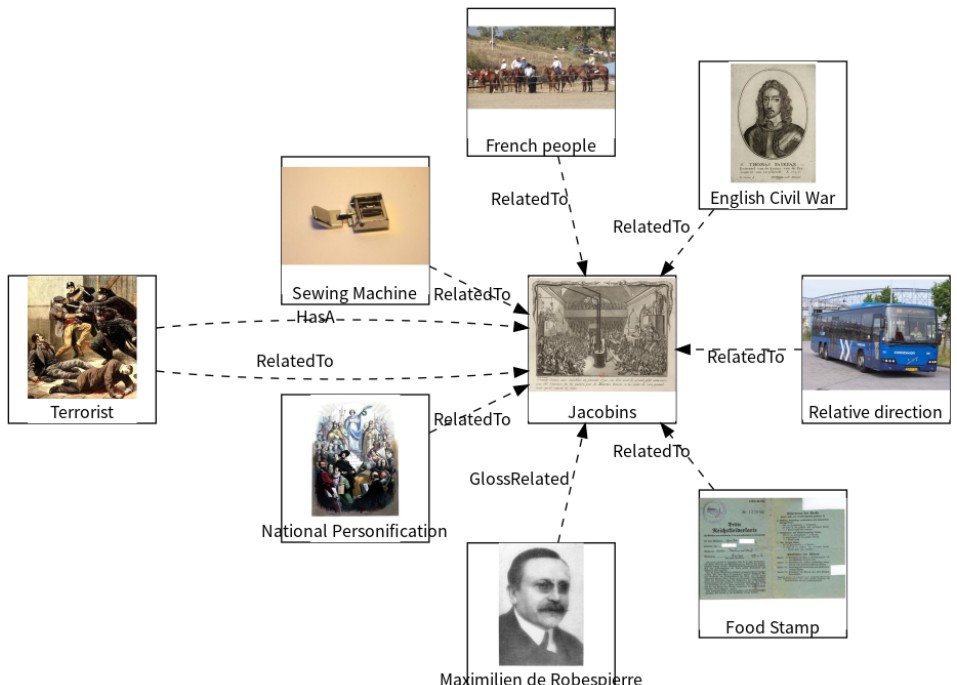

Figure 12: The MMKR image case for Task #2.

---

**Case for Task #2**

**Zero-shot Answer:**
The multi-modal knowledge graph in the image contains 10 different relations. These relations are:\n\n1. RelatedTo\n2. Submachinegun\n3. MachinePistol\n4. LightMachineGun\n5. AutomaticRifle\n6. FiringPortWeapon\n7. ColtAR15\n8. M201FiringPortWeapon\n9. GS30A1\n10. Heckler&KochG36"

**Stage 1 Answer:**
\n<think>\nThere are several different relations in the given multi-modal knowledge graph:\nRelatedTo\nGlossRelated\nTherefore, the number of different relations is 2\n</think>\n<answer>2</answer>\n

**Stage 2(DPO) Answer:**
\n<think>\nThere are several different relations in the given multi-modal knowledge graph:\nRelatedTo\nHasA\nGlossRelated\nTherefore, the number of different relations is 3\n</think>\n<answer>3</answer>\n

**Golden Answer:**
\n<think>\nThere are several different relations in the given multi-modal knowledge graph:\nRelatedTo\nIsA\nGlossRelated\nTherefore, the number of different relations is 3\n</think>\n<answer>3</answer>\n\n

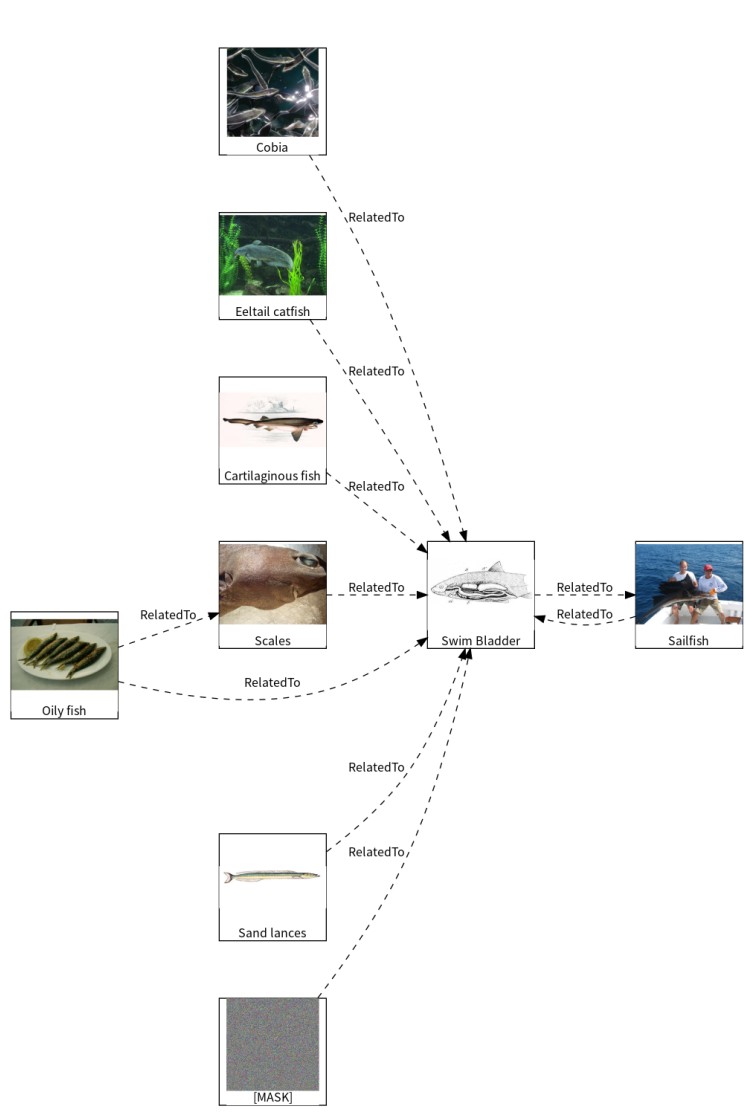

Figure 13: The MMKR image case for Task #7.

1026
1027
1028
1029
1030
1031
1032
1033
1034
1035
1036
1037
1038
1039
1040
1041
1042
1043
1044
1045
1046
1047
1048
1049
1050
1051
1052
1053
1054
1055
1056
1057
1058
1059
1060
1061
1062
1063
1064
1065
1066
1067
1068
1069
1070
1071
1072
1073
1074
1075
1076
1077
1078
1079

---

**Case for Task #7**

**Zero-shot Answer:**
To determine the correct entity that replaces the [MASK] in the given multi-modal knowledge graph, we need to analyze the relationships and the context provided by the other entities and their connections.\n\n1. The graph shows a hierarchy of relationships between different types of fish and their features.\n2. The [MASK] is connected to Scales and Swim Bladder, which are features of fish.\n3. The options provided are:\n - A. Tropical Africa\n - B. Carnivorous Fish\n - C. Bony Fish\n - D. Tropical fish\n - E. Philippine butterflyfish\n\nGiven that the [MASK] is related to Scales and Swim Bladder, it is likely to be a type of fish. Among the options, Carnivorous Fish and Tropical fish are types of fish, but Carnivorous Fish is a specific type of fish, while Tropical fish is a broader category that could include various types of fish.\n\nSince the graph is showing a hierarchy and the [MASK] is connected to Scales and Swim Bladder, it is more likely to be a specific type of fish rather than a broader category like Tropical fish. Therefore, the most appropriate choice is:\n\nB. Carnivorous Fish\n\nSo, the correct answer is B. Carnivorous Fish.

**Stage 1 Answer:**
\n<think>\nThe correct answer is Bony Fish because it is the most general and inclusive category that can logically relate to Swim Bladder across all the provided triples. The other options are either too specific (e.g., Tropical Africa, Tropical fish) or not directly relevant (e.g., Carnivorous Fish, Philippine butterflyfish). Bony Fish fits well with the general relationship to Swim Bladder and aligns with the context of the other fish-related entities in the triples.\n</think>\n<answer>C</answer>\n

**Stage 2(DPO) Answer:**
\n<think>\nThe correct entity Tropical fish is more suitable because it directly relates to the biological and ecological context of the Swim Bladder, which is a common feature in many fish species, including tropical fish. The other options are less suitable:\n\n- **A. Tropical Africa**: This is a geographical region and not a biological classification, making it irrelevant to the relationship with Swim Bladder.\n- **B. Carnivorous Fish**: While some fish are carnivorous, this option is too specific and does not encompass all fish that have a swim bladder.\n- **C. Bony Fish**: This is a broader classification but less specific than Tropical fish, which is a subset of bony fish that are commonly found in tropical waters.\n- **E. Philippine butterflyfish**: This is a specific species and not a general category, making it less suitable for the context of the triples.\n</think>\n<answer>D</answer>\n

**Golden Answer:**
\n<think>\nThe correct answer is Tropical fish (D) because it fits well within the context of the triples, which involve various types of fish and their relationship to the swim bladder. The other options are less suitable:\n\n- A. Tropical Africa: This is a region, not a type of fish, and does not fit the pattern of the triples.\n- B. Carnivorous Fish: While some carnivorous fish may have swim bladders, this term is too specific and doesn't encompass the variety of fish mentioned.\n- C. Bony Fish: This is a broader category that includes many fish with swim bladders, but Tropical fish is more specific and aligns better with the other fish types listed.\n- E. Philippine butterflyfish: This is a specific species, which is too narrow compared to the general category of Tropical fish.\n\nTropical fish is the most appropriate choice as it represents a diverse group of fish that often have swim bladders and fits well with the other fish types mentioned in the triples.\n</think>\n<answer>D</answer>\n\n

