# OpenReview forum: "Structured and Abstractive Reasoning on Multi-modal Relational Knowledge Images"
_ICLR.cc/2026/Conference — ICLR 2026 Conference Withdrawn Submission_

### Official Review · Reviewer_4nm1 · 2025-10-27

**Soundness:** 2
**Presentation:** 3
**Contribution:** 2
**Rating:** 4
**Confidence:** 3

**Summary:**

The paper introduces a method for automatically generating data to support multimodal relational knowledge and structured abstractive reasoning. Using this approach, the authors construct both a training dataset and a benchmark for evaluating MLLMs. The dataset consists of multimodal knowledge graphs along with questions spanning eight categories: entity counting, relation counting, image counting, triple counting, subgraph description, error detection, entity reasoning, and relation reasoning. To leverage this data, the authors propose a two-stage finetuning pipeline—supervised finetuning followed by preference alignment. Their experiments show that existing models perform poorly on the benchmark, while finetuning on the proposed dataset leads to substantial performance gains.

**Strengths:**

- The paper is well-written and clearly organized.
- Training on the proposed data leads to clear performance improvements.
- The analysis is thorough, exploring important dimensions such as task transferability, data scalability, and the effectiveness of chain-of-thought reasoning.

**Weaknesses:**

- The task appears somewhat artificial, with limited discussion of potential real-world applications. The experimental setup further highlights this artificiality: if I understand correctly, the visual format of the multimodal knowledge graphs remains fixed across all instances. Even if such a task were practically relevant, it is unlikely that real-world data would exhibit such consistent structure.
- Given that the dataset encodes factual information (e.g., concepts found in Wikipedia, a common pretraining source for LLMs), it would be useful to disentangle the contribution of visual input from the models’ reliance on prior textual knowledge. One way to validate this would be to include a baseline trained without image input (e.g., an additional row in Table 2 where only the question text is provided). Another option would be to construct counterfactual variants of the data and compare model performance under those conditions.

**Questions:**

- What are some concrete real-world scenarios where a multimodal knowledge graph of this kind would be useful?
- Have you evaluated the extent to which image information influences model predictions (see Weaknesses point 2)?
- Beyond the benchmark, what potential applications do these data have? Can the proposed method transfer to related tasks that use different schematics or alternative ways of representing relationships?
- What might explain the large performance differences between LLaVA1.5-Next and Qwen2.5VL? In the zero-shot setting, their performance is similar, with Qwen slightly ahead when controlling for model size. However, after training, this relationship reverses. What factors could be driving this change?

---

> ### Author Response · Authors · 2025-11-14
> **Author Response for Reviewer 4nm1**
>
> Dear Reviewer:
>
> Thank you for your insightful feedback. Below are the responses to the comments you raised.
>
> ### **Q1: Real-world Application Scenarios of STAR**
> In our design, we construct images with multi-modal relation knowledge (MMRK) and build a diverse benchmark to make the first step on MLLMs' capability exploration. This capability gives it a **distinct advantage in scenarios requiring multi-level reasoning and cross-modal alignment**. Some LLM research indicates that **training on tasks requiring logic and reasoning** such as mathematics and coding **can also enhance LLMs' capabilities across other general domains** [1]. Our study similarly explores whether training on MMRK images can improve the multimodal reasoning abilities of MLLMs, which is a general new data type for current MLLM community.
>
> Here, we'd like to present some classic application scenarios for MMRK images and STAR capability of MLLM:
>
> + Interactive learning in the education field. We can automatically converts textbook knowledge points (e.g., biological classification, historical event chains) into MMRK diagrams. Students trace knowledge pathways through Q&A interactions, while teachers assess comprehension gaps based on model reasoning processes.
> + Enterprise management applications. Internal corporate knowledge bases (such as product architecture and business processes) can be encoded as MMRK. Employees can obtain text-and-image answers by posing questions in natural language. The media industry can automatically generate relationship diagrams for news events.
> + E-commerce platforms can structure product attributes (brand, functionality, user reviews) into MMRK models, enabling users to perform precise searches using compound criteria (e.g., “portable devices suitable for outdoor activities”). In the cultural heritage sector, this approach can link artifact images, historical texts, and geographic information.
>
> In practice, this form of data finds application in numerous specific scenarios where structured knowledge manifests visually rather than textually. This places heightened demands on the STAR capabilities of MLLM models, and our research represents the first step in addressing this challenge.
>
>
>
> [1] DOES MATH REASONING IMPROVE GENERAL LLM CAPABILITIES? UNDERSTANDING TRANSFERABILITY OF LLM REASONING
>
> ### **Q2: The impact of Textual Knowledge in the LLM Backbone**
> We consider this suggestion valuable and believe it is necessary to test the performance of MLLM models that are entirely independent of visual information in ablation experiments. First, we need to declare some important points:
>
> + It is important to note that in our setup, all key information (MMRK) appears exclusively within the images. The text instruction component in the STAR benchmark consists solely of a series of task directives and does not contain any specific factual knowledge.
> + Our overarching goal is **not to explore how images can replace text to convey richer information**, **but rather to investigate how MLLMs can acquire the ability to understand and reason about structured knowledge from visual information when such knowledge exists solely within visual data. **Many tasks, such as counting in Tasks 1-4, become trivial when presented with textual information instead of images.
>
> We conducted the following experiment: We trained MLLMs on SFT using instruction datasets that excluded visual information, where the visual-based MMRK was represented as text prompts. The experimental settings and their corresponding results are as follows:
>
> + Trained and evaluated on original STAR. This is done in our submitted paper.
> + (Text-only -> Text-only) Trained and evaluated on text-only instructions (w/o MMRK images)
>
>
> + (Full -> Text-only) Trained on original STAR data (visual+text) and evaluated on text-only instructions
>
>
> + (Text-only -> Full) Trained on text-only instructions (w/o MMRK images) and evaluated on original STAR data (test set)

---

> > ### Author Response · Authors · 2025-11-14
> >
> > + Qwen2.5-VL-3B
> >
> > | Train Data | Test Data | Task1 | Task2 | Task3 | Task4 | Task5 | Task6 | Task7 | Task8 |
> > | --- | --- | --- | --- | --- | --- | --- | --- | --- | --- |
> > | Text-only | Text-only | 42.00 | 45.75 | 16.38 | 67.87 | 81.45 | 5.25 | 50.13 | 41.13 |
> > | Text-only | Full | 28.50 | 23.13 | 8.13 | 18.38 | 58.65 | 5.50 | 42.00 | 51.75 |
> > | Full | Text-only | 74.13 | 99.5 | 22.88 | 98.00 | 81.30 | 12.50 | 62.25 | 74.50 |
> >
> >
> > + Qwen2.5-VL-7B
> >
> > | Train Data | Test Data | Task1 | Task2 | Task3 | Task4 | Task5 | Task6 | Task7 | Task8 |
> > | --- | --- | --- | --- | --- | --- | --- | --- | --- | --- |
> > | Text-only | Text-only | 78.75 | 91.13 | 14.63 | 77.63 | 84.42 | 5.50 | 48.63 | 53.25 |
> > | Text-only | Full | 12.50 | 6.63 | 1.00 | 5.63 | 66.50 | 4.38 | 46.50 | 47.88 |
> > | Full | Text-only | 82.63 | 99.63 | 22.88 | 94.88 | 83.99 | 17.88 | 67.50 | 75.75 |
> >
> >
> > From these several experiments, we can make several additional interesting findings:
> >
> > 1. Understanding task would be easier if the MMRK is represented in the text form. We can find that MLLMs trained with text-only instructions still have good performance on Task 1,2,4,5, which needs to count the entity/relation/triple in the KG. Task 3 is an exception because it involves counting the number of images, which is clearly difficult to accomplish without visual information.
> > 2. Original STAR data provides strong generalization capability on text-only instructions. We found that models trained on raw multimodal instruction data maintain strong generalization capabilities even when processing pure text input, outperforming models trained solely on text instructions. This demonstrates that incorporating visual modalities into training yields significant performance gains for MLLMs.
> > 3. Training on text-only instructions fails to generalize capabilities to the visual modality, as evidenced by the significant performance degradation observed in the third set of supplementary experiments.
> >
> > Overall, visual information provides crucial generalization capabilities in MLLM, making its inclusion essential in our design.
> >
> >
> >
> > ### **Q3: Trasfer to Other Schematics of Relationship Representing**
> > In Q1, we have already listed numerous specific application scenarios for MMRK data. We believe the same approach can be extended to different relationship representation formats. However, the current focus of this paper remains on knowledge graphs—that is, using graph structures to represent complex and abstract relations. We will also contribute more insights in this area in the future.
> >
> > ### **Q4: Performance Different Between Qwen and LLaVA**
> > It is worth noting that in our experiments, the Qwen series models uniformly employed Qwen2.5-VL, while the LLaVA models utilized two variants: LLaVA-1.5-7B and LLaVA-NEXT-34B. When comparing LLaVA-1.5-7B to the similarly sized Qwen2.5-VL-7B, the former demonstrated overall poorer performance—both in zero-shot scenarios and after training. When comparing LLaVA-NEXT-34B and Qwen2.5-VL-32B, LLaVA-NEXT-34B performed worse in zero-shot scenarios but significantly outperformed Qwen2.5-VL-32B after training.
> >
> > Overall, the reason for this phenomenon is that Qwen models across different sizes exhibit less overall fluctuation in performance on the STAR dataset, but also have a lower upper bound. In contrast, LLaVA series models achieve a higher upper bound on these tasks. This discrepancy may stem from various factors, including differences in architectural design, corpus selection, and training methods between the two model families.

---

> > > ### Author Response · Authors · 2025-11-14
> > >
> > > ### **PS: Additional GRPO Training Strategy in Stage2 is Reported**
> > > Considering that RL-based methods such as GRPO are also highly effective MLLM capability enhancement approaches, we incorporated GRPO into stage 2 training. This differs from other alignment methods (DPO/ORPO/SimPO) used in the original paper, and the corresponding results are as follows:
> > >
> > > | Setting | | Task1 | Task2 | Task3 | Task4 | Task5 | Task6 | Task7 | Task8 | Overall |
> > > | --- | --- | --- | --- | --- | --- | --- | --- | --- | --- | --- |
> > > | Qwen2.5-VL-3B | w/ DPO | 55.50  | 89.25  | 66.88  | 26.13  | 66.64  | 37.50  | 60.00  | 76.85  | 59.84  |
> > > | | w/ ORPO | 39.00  | 84.62  | 59.00  | 17.88  | 66.81  | 37.75  | 59.63  | 77.63  | 55.29  |
> > > | | w/ SimPO | 71.00  | 89.38  | 37.25  | 28.13  | 67.92  | 37.62  | 59.13  | 78.25  | 58.59  |
> > > | | w/ GRPO | 65.75  | 84.88  | 62.38  | 16.50  | 71.40  | 43.87  | 64.13  | 79.75  | 61.08  |
> > > | Qwen2.5-VL-7B | w/ DPO | 66.50  | 94.00  | 73.50  | 30.25  | 76.44  | 58.63  | 69.37  | 82.00  | 68.84  |
> > > | | w/ ORPO | 65.75  | 93.38  | 71.88  | 27.38  | 76.65  | 56.38  | 70.00  | 79.75  | 67.65  |
> > > | | w/ SimPO | 69.63  | 93.75  | 75.38  | 29.00  | 76.32  | 57.75  | 68.50  | 81.50  | 68.98  |
> > > | | w/ GRPO | 71.38  | 93.50  | 71.13  | 27.63  | 77.92  | 57.88  | 71.88  | 80.88  | 69.03  |
> > >
> > >
> > > + In our GRPO implementation, we use the accuracy of the final prediction result as the basis for calculating the reward, employing a 0-1 reward system. If the answer is correct, the corresponding reward is 1; otherwise, it is 0.
> > > + We use the same datasets used by DPO/ORPO/SimPO.
> > >
> > > From the additional experimental results, we can observe that GRPO achieves better results compared with other alignment methods in S2.
> > >
> > >
> > >
> > > Thank you! We hope that you can consider raise up your score based on our response.

---

> > > > ### Comment · Reviewer_4nm1 · 2025-11-27
> > > >
> > > > Thank you for the response. I will keep my current score, as I am still unsure how the present data format (which lacks structural variations) meaningfully maps to real-world domains. In addition, the issue of separating prior LLM knowledge from the MMKG input is not fully addressed in either the rebuttal or the paper.
> > > > While I appreciate the modality-ablation experiments, they do not directly target the core question. My original intention was to evaluate a setup where the entire KG context is removed, leaving only the question text, rather than removing individual modalities.

---

### Official Review · Reviewer_RqkB · 2025-10-27

**Soundness:** 3
**Presentation:** 3
**Contribution:** 2
**Rating:** 4
**Confidence:** 4

**Summary:**

This paper investigates the problem of structured and abstractive reasoning over multimodal relational knowledge in MLLMs. The authors developed a data pipeline to construct the MMRK dataset and related evaluation strategies. Based on Qwen2.5-VL and LLaVA-Next, they conducted in-depth experiments, analyzing training strategies and the characteristics of models trained on the MMRK data.

**Strengths:**

* The authors built a comprehensive data pipeline that covers data source collection, data/instruction/task synthesis, as well as data validation strategies.
* The authors conducted an in-depth exploration of the impact of training strategies on learning MMRK, involving two different mainstream multimodal foundation models and multiple post-training approaches. The experimental results are highly insightful.
* Moreover, the small models trained on the authors’ synthesized data can achieve performance comparable to, or even better than, that of large closed-source models, further confirming the scarcity of data in this field and the significance of the dataset constructed in this work.

**Weaknesses:**

* The authors’ experiments are mainly centered on in-domain tasks that are similar to the training data, lacking evaluation on real-world problems. As shown in Figure 1, the authors present a variety of scenarios. Experimental results on these scenarios would help to better understand the impact of the authors’ contribution on the model’s real-world performance.
* The authors use an LLM (Qwen2.5-VL-72B) as the judge, but its reliability has not been thoroughly discussed. This is especially noteworthy given that the zero-shot Qwen2.5-VL-32B demonstrates performance comparable to that of Qwen2.5-VL-72B.
* The authors have thoroughly demonstrated the performance improvements brought by the MMRK data. But the baseline models compared by the authors are relatively outdated, and they did not include comparisons with the latest models such as Qwen3-VL, Claude 4.5, or Gemini 2.5 Pro. Comparing against state-of-the-art models would help readers better understand the current capabilities of multimodal models in solving STAR tasks.

**Questions:**

refer to the weaknesses

---

> ### Author Response · Authors · 2025-11-14
> **Author Response for Reviewer RqkB**
>
> Dear Reviewer:
>
> Thank you for your insightful feedback. Below are the responses to the comments you raised.
>
> ### **Q1: More Results on Real-world Problems**
> Your consideration of this point is highly valuable, so we conducted further experiments on additional real-world benchmarks. Below are the results from testing Qwen2.5-VL-7B on the MMMU benchmark—one of the most widely adopted MLLM general multimodal capability benchmarks today. We evaluated the performance of MLLMs at different stages on MMMU.
>
> | Domain | Art & Design | Business | Science | Health & Medicine | Humanities & Social Science | Tech & Engineering | Overall |
> | --- | --- | --- | --- | --- | --- | --- | --- |
> | Baseline | 14.28 | 21.83 | 25.45 | 10.40 | 8.57 | 9.81 | 15.06 |
> | S1(SFT) | 15.23 | 28.25 | 25.31 | 12.92 | 9.04 | 12.50 | 17.20 |
> | S2(DPO) | 15.95 | 28.77 | 24.32 | 12.21 | 9.16 | 12.31 | 17.12 |
> | S2(ORPO) | 15.47 | 29.21 | 24.41 | 12.40 | 9.15 | 11.63 | 17.05 |
> | S2(SimPO) | 14.76 | 25.55 | 23.92 | 12.22 | 9.04 | 11.06 | 16.09 |
>
>
>
>
> We can ovserve an interesting phenomenon: after training on our STAR benchmark, the model generally achieves performance gains across various general domains in the MMMU. Notably, the data distribution of the STAR benchmark is markedly different from that of the MMMU. This suggests that after two-stage training, certain capabilities of the MLLM have been activated rather than experiencing catastrophic forgetting. It also indicates that the STAR data does not conflict with the existing MLLM training corpus, thereby preventing the model from catastrophic forgetting.
>
> We also tried common capability evaluation on OCRBench [1] and TextVQA [2], which are two popular benchmarks used by current MLLM evaluation. The results are presented below:
>
> | Domain | OCRBench | TextVQA |
> | --- | --- | --- |
> | Baseline | 61.7 | 52.32 |
> | S1(SFT) | 61.3 | 54.28 |
> | S2(DPO) | 62.4 | 54.52 |
> | S2(ORPO) | 62.3 | 54.36 |
> | S2(SimPO) | 62.0 | 54.58 |
>
>
> The results show that after training on the STAR dataset, the MLLM did not experience catastrophic forgetting on these two benchmarks but instead demonstrated performance improvements. This indicates that our data and training method design are harmless to the general ability of MLLMs.
>
> ### **Q2: Reliability of the Evaluation Metrics**
> We'd like to elaborate on our evaluation protocol. We assess tasks in two parts: the accuracy of the final answer and the quality of the CoT. For the accuracy component, we do not use LLM scoring. Instead, we directly extract the model-generated answer from the structured CoT and compare it against the correct answer to compute the result. For the CoT component, we employ LLM as the scorer, following the classic LLM-as-a-judge paradigm.
>
> Therefore, our approach combines rule-based methods with LLM-based evaluation to provide a more comprehensive reflection of model performance. Regarding CoT accuracy comparisons, most current approaches also employ LLMs as scorers. We provide a reference answer, allowing the LLM to compare the similarity between its prediction and the standard answer. This process utilizes the Qwen2.5-72B model rather than the VL variant, focusing solely on textual similarity without incorporating visual information. Under these conditions, the LLM scorer we employ is considered reliable.
>
> ### **Q3: More Baseline Results on the STAR data**
> This feedback is highly valuable. However, prior to the ICLR submission deadline, certain models such as Qwen3-VL were not publicly available. We now provide more comprehensive baseline results as follows:
>
> | Model | Task1 | Task2 | Task3 | Task4 | Task5 | Task6 | Task7 | Task8 | Overall |
> | --- | --- | --- | --- | --- | --- | --- | --- | --- | --- |
> | Qwen3-VL-30B | 94.25 | 93.75 | 7.88 | 62.13 | 57.63 | 0.5 | 8.63 | 41.00 | 45.72 |
>
>
>
>
>
>
>
> Thank you! We hope that you can consider raise up your score based on our response.
>
>
>
> [1] OCRBench: On the Hidden Mystery of OCR in Large Multimodal Models
>
> [2] Towards vqa models that can read

---

> > ### Author Response · Authors · 2025-11-14
> > **Additional Response**
> >
> > ### **PS: Additional GRPO Training Strategy in Stage2 is Reported**
> > Considering that RL-based methods such as GRPO are also highly effective MLLM capability enhancement approaches, we incorporated GRPO into stage 2 training. This differs from other alignment methods (DPO/ORPO/SimPO) used in the original paper, and the corresponding results are as follows:
> >
> > | Setting | | Task1 | Task2 | Task3 | Task4 | Task5 | Task6 | Task7 | Task8 | Overall |
> > | --- | --- | --- | --- | --- | --- | --- | --- | --- | --- | --- |
> > | Qwen2.5-VL-3B | w/ DPO | 55.50  | 89.25  | 66.88  | 26.13  | 66.64  | 37.50  | 60.00  | 76.85  | 59.84  |
> > | | w/ ORPO | 39.00  | 84.62  | 59.00  | 17.88  | 66.81  | 37.75  | 59.63  | 77.63  | 55.29  |
> > | | w/ SimPO | 71.00  | 89.38  | 37.25  | 28.13  | 67.92  | 37.62  | 59.13  | 78.25  | 58.59  |
> > | | w/ GRPO | 65.75  | 84.88  | 62.38  | 16.50  | 71.40  | 43.87  | 64.13  | 79.75  | 61.08  |
> > | Qwen2.5-VL-7B | w/ DPO | 66.50  | 94.00  | 73.50  | 30.25  | 76.44  | 58.63  | 69.37  | 82.00  | 68.84  |
> > | | w/ ORPO | 65.75  | 93.38  | 71.88  | 27.38  | 76.65  | 56.38  | 70.00  | 79.75  | 67.65  |
> > | | w/ SimPO | 69.63  | 93.75  | 75.38  | 29.00  | 76.32  | 57.75  | 68.50  | 81.50  | 68.98  |
> > | | w/ GRPO | 71.38  | 93.50  | 71.13  | 27.63  | 77.92  | 57.88  | 71.88  | 80.88  | 69.03  |
> >
> >
> > + In our GRPO implementation, we use the accuracy of the final prediction result as the basis for calculating the reward, employing a 0-1 reward system. If the answer is correct, the corresponding reward is 1; otherwise, it is 0.
> > + We use the same datasets used by DPO/ORPO/SimPO.
> >
> > From the additional experimental results, we can observe that GRPO achieves better results compared with other alignment methods in S2.
> >
> >
> >
> > Thank you! We hope that you can consider raise up your score based on our response.
> >
> >
> >
> > [1] OCRBench: On the Hidden Mystery of OCR in Large Multimodal Models
> >
> > [2] Towards vqa models that can read

---

> ### Comment · Reviewer_RqkB · 2025-11-28
>
> Thank you for your response. I will keep my current score, as it is difficult to see clear evidence that the STAR data contributes to solving real-world problems or benchmarks. The improvements on OCRBench and TextVQA are quite limited, and I also noticed that the baseline performance is much lower than what was reported by the Qwen team.

---

### Official Review · Reviewer_ZxNE · 2025-10-29

**Soundness:** 2
**Presentation:** 3
**Contribution:** 3
**Rating:** 4
**Confidence:** 5

**Summary:**

This paper tackles Structured and Abstractive Reasoning (STAR) with Multi-Modal Relational Knowledge (MMRK), an underexplored area for MLLMs. Key contributions include an automatic STAR data engine for multi-modal instruction synthesis and a two-stage enhancement framework. Introducing the **STAR-64K** dataset, experiments show smaller 3B/7B models outperform GPT-4o in STAR tasks. The work provides valuable insights into model performance, data scalability, and transferability.

**Strengths:**

1.**Clear Motivation**: The paper presents a well-defined objective, introducing a data engine to synthesize Multi-Modal Relational Knowledge (MMRK) data. Training on this data effectively enhances models' abilities in Structured and Abstractive Reasoning (STAR).

2.**Novel Contribution**: This is the first work to transform multi-modal knowledge graphs into reasoning tasks involving multi-modal relational knowledge. The authors design training data, evaluation benchmarks, and validate the approach on 3B/7B models, addressing a previously unexplored gap in research.

**Weaknesses:**

While incorporating STAR training effectively enhances multi-modal models' structured and abstractive reasoning capabilities, it remains unclear whether other abilities, such as OCR (TextVQA,DocVQA)and general multi-modal question answering (MME,MMMU), may suffer from catastrophic forgetting. If the improvement in STAR comes at the expense of reducing the model's general versatility or impairing core functions like OCR, the practical significance of the framework would be greatly diminished. Further exploration into the trade-offs between STAR enhancement and general multi-modal model performance is necessary to validate its broader applicability and impact.

**Questions:**

The key question is whether the addition of STAR capabilities is genuinely impactful or simply included for novelty. It is critical to establish whether integrating STAR tasks enhances the general versatility and performance of multi-modal models. To address this, I strongly recommend the authors conduct an ablation study comparing the performance of a model trained with and without STAR data on comprehensive multi-modal benchmarks, using a baseline such as the LLAVA framework. This experiment would demonstrate the necessity and real-world significance of STAR tasks.

If this concern is adequately addressed and the results prove STAR's contribution to improving general multi-modal reasoning, I will consider revising my score positively.

---

> ### Author Response · Authors · 2025-11-14
> **Author Response for Reviewer ZxNE**
>
> Dear Reviewer:
>
> Thank you for your insightful feedback. Below are the responses to the comments you raised.
>
> ### **Q1: General Ability of MLLMs After Training on the STAR data**
> The consideration you mentioned is crucial, and it is precisely what we will address in our paper. We believe training on MMRK data enables LLMs to maintain strong performance on general-purpose benchmarks. Specifically, we conducted benchmark evaluations of MMMU using Qwen2.5-VL-7B, comparing the performance of different training stages (base model, model after S1, and model after S2) on this multimodal common-sense knowledge benchmark. The results are as follows:
>
> | Domain | Art & Design | Business | Science | Health & Medicine | Humanities & Social Science | Tech & Engineering | Overall |
> | --- | --- | --- | --- | --- | --- | --- | --- |
> | Baseline | 14.28 | 21.83 | 25.45 | 10.40 | 8.57 | 9.81 | 15.06 |
> | S1(SFT) | 15.23 | 28.25 | 25.31 | 12.92 | 9.04 | 12.50 | 17.20 |
> | S2(DPO) | 15.95 | 28.77 | 24.32 | 12.21 | 9.16 | 12.31 | 17.12 |
> | S2(ORPO) | 15.47 | 29.21 | 24.41 | 12.40 | 9.15 | 11.63 | 17.05 |
> | S2(SimPO) | 14.76 | 25.55 | 23.92 | 12.22 | 9.04 | 11.06 | 16.09 |
>
>
> An interesting phenomenon can be observed: after training on our synthetic STAR benchmark, the model generally achieves performance gains across various general domains in the MMMU. Notably, the data distribution of the STAR benchmark is markedly different from that of the MMMU. This suggests that after two-stage training, certain capabilities of the MLLM have been activated rather than experiencing catastrophic forgetting. It also indicates that the STAR data does not conflict with the existing MLLM training corpus, thereby preventing the model from from catastrophic forgetting.
>
> We also tried common capability evaluation on OCRBench and TextVQA, which are two popular benchmarks used by current MLLM evaluation. The results are presented below:
>
> | Domain | OCRBench | TextVQA |
> | --- | --- | --- |
> | Baseline | 61.7 | 52.32 |
> | S1(SFT) | 61.3 | 54.28 |
> | S2(DPO) | 62.4 | 54.52 |
> | S2(ORPO) | 62.3 | 54.36 |
> | S2(SimPO) | 62.0 | 54.58 |
>
>
> The results show that after training on the STAR dataset, the MLLM did not experience catastrophic forgetting on these two benchmarks but instead demonstrated performance improvements. This indicates that our data and training method design are harmless to the general ability of MLLMs.
>
>
>
> ### **PS: Additional GRPO Training Strategy in Stage2 is Reported**
> Considering that RL-based methods such as GRPO are also highly effective MLLM capability enhancement approaches, we incorporated GRPO into stage 2 training. This differs from other alignment methods (DPO/ORPO/SimPO) used in the original paper, and the corresponding results are as follows:
>
> | Setting | | Task1 | Task2 | Task3 | Task4 | Task5 | Task6 | Task7 | Task8 | Overall |
> | --- | --- | --- | --- | --- | --- | --- | --- | --- | --- | --- |
> | Qwen2.5-VL-3B | w/ DPO | 55.50  | 89.25  | 66.88  | 26.13  | 66.64  | 37.50  | 60.00  | 76.85  | 59.84  |
> | | w/ ORPO | 39.00  | 84.62  | 59.00  | 17.88  | 66.81  | 37.75  | 59.63  | 77.63  | 55.29  |
> | | w/ SimPO | 71.00  | 89.38  | 37.25  | 28.13  | 67.92  | 37.62  | 59.13  | 78.25  | 58.59  |
> | | w/ GRPO | 65.75  | 84.88  | 62.38  | 16.50  | 71.40  | 43.87  | 64.13  | 79.75  | 61.08  |
> | Qwen2.5-VL-7B | w/ DPO | 66.50  | 94.00  | 73.50  | 30.25  | 76.44  | 58.63  | 69.37  | 82.00  | 68.84  |
> | | w/ ORPO | 65.75  | 93.38  | 71.88  | 27.38  | 76.65  | 56.38  | 70.00  | 79.75  | 67.65  |
> | | w/ SimPO | 69.63  | 93.75  | 75.38  | 29.00  | 76.32  | 57.75  | 68.50  | 81.50  | 68.98  |
> | | w/ GRPO | 71.38  | 93.50  | 71.13  | 27.63  | 77.92  | 57.88  | 71.88  | 80.88  | 69.03  |
>
>
> + In our GRPO implementation, we use the accuracy of the final prediction result as the basis for calculating the reward, employing a 0-1 reward system. If the answer is correct, the corresponding reward is 1; otherwise, it is 0.
> + We use the same datasets used by DPO/ORPO/SimPO.
>
> From the additional experimental results, we can observe that GRPO achieves better results compared with other alignment methods in S2.
>
> We will continue to optimize the method design of our paper to make more technical contribution.
>
>
>
> Thank you! We hope that you can consider raise up your score based on our response.

---

### Author Response · Authors · 2025-11-14
**A Gentle Message to All Reviewers to Read our Response**

Dear reviewers:

We have addressed each and every question raised by everyone. The main newly added experimental results include:

- General ability (MMMU, OCRBench, TextVQA) of MLLMs after training on the STAR benchmark.
- Ablation study on training without any visual inputs and cross-modality generalization.
- New S2 strategy training MLLMs' STAR capability with GRPO.

**We hope the reviewers will review our response and consider raising the score. I believe our work and the additional rebuttal experiments can provide significant insights.** We will further upload the revised paper soon. Thank you!

Sincerely,
Authors

---

> ### Author Response · Authors · 2025-11-17
> **A Gentle Message to Remind Reviewers for Rebuttal Participation**
>
> Dear Reviewers:
>
> We've entered a new week, and we hope you can participate in the rebuttal session and provide further feedback. Thank you!
>
> Sincerely,
> Authors

---

### Note · Authors · 2025-12-30

I have read and agree with the venue's withdrawal policy on behalf of myself and my co-authors.